# Familial Occurrence of Adult Granulosa Cell Tumors: Analysis of Whole-Genome Germline Variants

**DOI:** 10.3390/cancers13102430

**Published:** 2021-05-18

**Authors:** Joline F. Roze, Joachim Kutzera, Wouter Koole, Margreet G. E. M. Ausems, Kristi Engelstad, Jurgen M. J. Piek, Cor D. de Kroon, René H. M. Verheijen, Gijs van Haaften, Ronald P. Zweemer, Glen R. Monroe

**Affiliations:** 1Department of Gynaecological Oncology, UMC Utrecht Cancer Center, University Medical Center Utrecht, Utrecht University, 3584 CX Utrecht, The Netherlands; rene.h.m.verheijen@gmail.com (R.H.M.V.); Gmonroe1@its.jnj.com (G.R.M.); 2Department of Genetics, University Medical Center Utrecht, Utrecht University, 3584 CX Utrecht, The Netherlands; J.Kutzera@umcutrecht.nl (J.K.); W.Koole@umcutrecht.nl (W.K.); M.G.E.M.Ausems@umcutrecht.nl (M.G.E.M.A.); G.vanHaaften@umcutrecht.nl (G.v.H.); 3FM Ambulance, Fargo, ND 58078, USA; kristi.engelstad@fmambulance.com; 4Department of Obstetrics and Gynaecology, Catharina Hospital, 5623 EJ Eindhoven, The Netherlands; jurgen.piek@catharinaziekenhuis.nl; 5Department of Obstetrics and Gynaecology, Leiden University Medical Center, 2333 ZA Leiden, The Netherlands; C.D.de_Kroon@lumc.nl

**Keywords:** granulosa cell tumor, sex cord-stromal tumors, ovarian cancer, whole-genome sequencing, *FOXL2*

## Abstract

**Simple Summary:**

Although granulosa cell tumors can occur in rare syndromes and one familial case of a granulosa cell tumor has been described, a genetic predisposition for granulosa cell tumors has not been identified. Through our collaborations with patients, we identified four families in which two women of each family were diagnosed with an adult granulosa cell tumor. Although predicted deleterious variants in *PIK3C2G*, *BMP5,* and *LRP2* were found, we could not identify an overlapping genetic variant or affected locus that may explain a genetic predisposition for granulosa cell tumors. The age of onset in the familial patients was significantly lower (median 38 years, range from 17 to 60) than in sporadic patients (median between 50 and 55 years). Furthermore, breast cancer, polycystic ovary syndrome, and subfertility were seen in these families.

**Abstract:**

Adult granulosa cell tumor (AGCT) is a rare ovarian cancer subtype, with a peak incidence around 50–55 years. Although AGCT can occur in specific syndromes, a genetic predisposition for AGCT has not been identified. The aim of this study is to identify a genetic variant in families with AGCT patients, potentially contributing to tumor evolution. We identified four families, each including two women diagnosed with AGCT. Whole-genome sequencing was performed to identify overlapping germline variants or affected genes. Familial relationship was evaluated using genealogy and genomic analyses. Patient characteristics, medical (family) history, and pedigrees were collected. Findings were compared to a reference group of 33 unrelated AGCT patients. Mean age at diagnosis was 38 years (range from 17 to 60) versus 51 years in the reference group, and seven of eight patients were premenopausal. In two families, three first degree relatives were diagnosed with breast cancer. Furthermore, polycystic ovary syndrome (PCOS) and subfertility was reported in three families. Predicted deleterious variants in PIK3C2G, BMP5, and LRP2 were identified. In conclusion, AGCTs occur in families and could potentially be hereditary. In these families, the age of AGCT diagnosis is lower and cases of breast cancer, PCOS, and subfertility are present. We could not identify an overlapping genetic variant or affected locus that may explain a genetic predisposition for AGCT.

## 1. Introduction

Ovarian cancer is the fifth leading cause of cancer-related death among women and arises from epithelial, sex-cord stromal or germ cells [1]. Granulosa cell tumors belong to the sex-cord stromal tumors and represent 5% of ovarian cancers, with an estimated incidence of 0.6–1.0 in 100,000 women worldwide per year [2]. The tumor is derived from the hormonally active granulosa cells that produce estradiol. Patients may develop symptoms, such as vaginal bleeding, caused by prolonged estrogen exposure or abdominal pain. Occasionally, a granulosa cell tumor is diagnosed in patients presenting with subfertility, potentially as a result of unregulated inhibin secretion by the tumor [3,4]. Although granulosa cell tumors can occur at any age, they mostly present perimenopausal or early in postmenopause, with a median age of diagnosis between 50 and 54 years [2]. Granulosa cell tumors are subdivided into an adult type (95%) and juvenile type (5%) by their histological and molecular characteristics [5], the latter mainly occurring at a younger age. 

Germline mutations are involved in the evolution of some ovarian cancers. Approximately 10–15% of epithelial ovarian cancer is caused by a germline mutation in the *BRCA1* or *BRCA2* gene [6]. In addition, specific hereditary syndromes result in an increased risk for sex cord-stromal tumors. Peutz–Jeghers syndrome is a rare autosomal dominant disease caused by germ line mutations in *STK11/LKB1*. Mutations in this gene are associated with gastrointestinal polyps, pigmentation of lips, and ovarian granulosa, and Sertoli cell tumors [7]. In addition, a germline mutation in *DICER1* can cause a hereditary syndrome that is associated with Sertoli–Leydig cell tumors [8]. Furthermore, Olliers disease and Maffucci syndrome are rare disorders caused by early post-zygotic mutations in *IDH1* and *IDH2* genes and are associated with juvenile granulosa cell tumors [9]. 

Adult granulosa cell tumors (AGCTs) harbor a specific somatic *FOXL2* c.402C > G mutation in approximately 95% of cases [10]. The C134W protein change caused by this mutation leads to reduced apoptosis, although the mechanism of granulosa cell tumorigenesis has not yet been entirely unraveled [11]. *FOXL2* is preferentially expressed in the ovary, the eyelids, and the pituitary gland. Inactivating germline mutations in this gene do not result in AGCTs but in the autosomal dominant blepharophimosis–ptosis–epicanthus inversus syndrome (BPES) type I [12]. This disease affects the eyelids and is associated with granulosa cell dysfunction and premature ovarian failure.

Despite the fact that rare autosomal disorders are associated with the development of granulosa cell tumors, there is no known genetic predisposition that is specific for AGCT. To date, there has been only one reported case of a family in which both mother and daughter were diagnosed with a granulosa cell tumor [13]. This was seen as a coincidental finding and DNA analysis was not performed. We identified four different families in which two relatives were diagnosed with a granulosa cell tumor.

In this study, we performed whole-genome sequencing on familial AGCT patients to investigate overlapping germline mutations or shared affected genes as a potential cause for AGCT development. Findings were compared to a reference group of 33 unrelated AGCT patients [14]. Identification of an overlapping germline variant or affected genetic locus in these families could help to unravel the pathological mechanism of tumor evolution in AGCT patients.

## 2. Materials and Methods

We identified four different families that each had two women previously diagnosed with an AGCT. Patients were identified through contact with the national and international granulosa cell tumor patient organization and gynecological oncologists involved in our national research on granulosa cell tumors [15]. Ethical approval was obtained from the Institutional Review Board of the University Medical Center Utrecht (UMCU METC 17–868). All participants provided written informed consent. Peripheral blood samples *(n* = 6) or saliva swabs *(n* = 2) were collected for germline DNA analysis. DNA isolation and sequencing was performed according to previously described methods [14]. Clinical data were provided by the treating gynecologists *(n* = 6) or by the patients themselves *(n* = 2), including the age at diagnosis, disease stage, medical history, and family history. We retrieved information on the family relations and on the medical history of non-affected family members, in order to build pedigrees. A genealogist traced back family lineages to investigate whether a relationship between the four families existed. In addition, we used TRIBES to assess genetic relatedness within and between families. TRIBES is a pipeline for relatedness detection in genomic data, using the 1000 Genomes European cohort, which can accurately assess genetic relatedness up to 7th degree relatives [16]. Additionally, we used the whole-genome sequencing results of germline DNA from 33 unrelated AGCT patients as a reference group [14]. We tested for potential relatedness within and between the reference group and the families. Furthermore, we investigated whether clinical characteristics (i.e., age of onset and disease stage) differed between the related AGCT patients and the unrelated reference group and checked whether overlapping germline variants or affected genes present in the germline DNA of the families, were also present in the germline DNA of the reference group.

### 2.1. Whole-Genome Sequencing and Variant Calling

Whole-genome sequencing was performed with 30× coverage on Illumina HiSeq X or NovaSeq 6000 instrument (Illumina, San Diego, CA, USA) by the Hartwig Medical Foundation (HMF, Amsterdam, NL, USA). Genome analysis was performed using the UMC IAP pipeline [17]. Sequence reads were mapped with Burrows–Wheeler Alignment v0.75a against human reference genome GRCh37. Single nucleotide variants and small insertions and deletions were called with GATK (v3.4.46). The functional effect of these variants was predicted with SnpEff (v.4.1). Structural variants were called using DELLY (0.7.2) [18] and Manta(v0.29.5).

### 2.2. Genome Analysis

First, all coding variants shared *within* families were analyzed. Variants with a high population frequency (>1:1000 according to population databases [19,20,21,22]) were filtered out, as AGCT constitutes a rare malignancy. We further removed frequent variants and sequencing artefacts using an in-house cancer reference database from the Hartwig Medical Foundation (HMF pool of normal variants V2.0, a resource with 78655034 unique variant calls from 1762 individuals) [23]. The exonic (± 10 base pairs into the intronic regions) variant analysis was performed using Alissa (Agilent Technologies Alissa Interpret v5.1.7). Furthermore, variants present in the COSMIC Cancer Gene Census (release v89) were annotated and reviewed (Appendix A). We assessed all variants present in cancer genes in the Human Gene Mutation Database (HGMD^®^ Professional 2020.3) to identify potential inherited disease-causing mutations. Finally, variants genome-wide were ranked on their potential pathogenicity by using the Combined Annotation Dependent Depletion (CADD PHRED) score [24]. Variants with a deleteriousness score ≥ 5 were annotated (Appendix A), and intra-family variants with a score ≥ 20 were assessed.

Second, we investigated overlapping genome-wide variants and affected genes between families. To assess shared variants, we selected variants present in both individuals in at least one family, and all variants shared by ≥2 families were evaluated by predicted pathogenicity of the variant and gene function. Additionally, we searched for recurrently affected genes across different families and evaluated them on mutation effect and gene function.

Third, we investigated larger structural variations across samples. Structural variants present in the GNOMAD catalogue [25] and the HMF SV pool of normals were removed prior to analysis. We used in-house scripts based on the R-package StructuralVariantAnnotation to analyze the samples [26]. Shared breakpoints were determined using the function “findBreakpointOverlaps” from StructuralVariantAnnotation with default parameters, and variants with overlapping breakpoints between samples were grouped together. Structural variants present in less than one family were removed. We annotated the remaining structural variants, the genes present within each structural variant region and investigated those genes for known involvement in granulosa cells or cancer development (Appendix A).

## 3. Results

### 3.1. Description of Families

We identified four families with two women diagnosed with AGCT, three originating from the Netherlands and one American family. Pedigrees are shown in Figure 1. Family lineages were traced back for at least seven generations. All families (for 7–9 generations) came from different geographical areas and did not have any overlapping family names. Furthermore, their genomic relatedness correlated directly with their actual relatedness based on the pedigrees and there was no close genetic relationship between families (all predicted ≥ 9th degree relatedness according to TRIBES, see Appendix A). None of the families were linked, indicating families did not share a recent common ancestor.

Family A included two women with an AGCT, who were fifth-degree relatives as their grandfathers were brothers. The patients were diagnosed with stage IIB and IA AGCT at 35 and 36 years of age, respectively. The medical history revealed congenital clubfoot in the first patient and subfertility associated with polycystic ovary syndrome in the other patient (Table 1). Family B contained two first-degree relatives, mother and daughter, both diagnosed with AGCT at the age of 39 and 46, respectively. The mother underwent surgery and radiotherapy for stage IC disease and the daughter was treated with surgery only for stage IC AGCT. The five sisters and three brothers of the affected mother, as well as their offspring (in total 19 children), were all unaffected by any cancer type. Family C included two second-degree relatives (aunt and niece) diagnosed with stage IA AGCT at 35 and 60 years of age, respectively. The first patient was diagnosed after unsuccessful subfertility treatment, as she suffered from polycystic ovary syndrome. More cases of subfertility were reported in this family. The respective mother and sister of the affected patients was diagnosed with breast cancer when she was 56 years old. Family D consisted of two fifth-degree relatives, related via their grandfathers, diagnosed with stage IC AGCT at the age of 17 and 39 years. The mothers of the affected patients had a history of breast cancer and were diagnosed at 38 and 61 years of age. Furthermore, one of the patient’s sisters was unable to conceive. All patients were successfully treated with surgery *(n* = 5), with surgery and radiotherapy *(n* = 1), or with surgery and chemotherapy *(n* = 2). None of the patients developed a recurrence during a median follow up time of 6 years (range from 4 to 38 years). In summary, polycystic ovary syndrome (PCOS) and subfertility were reported in three families and three first degree relatives had a history of breast cancer. 

### 3.2. Exome Analysis

Within the families, 26 to 202 shared coding germline variants (exonic +/−10 base pairs) were detected (Figure 2). Exonic variants passing quality filters that resulted in a frameshift, stop/start loss, or a nonsynonymous variant predicted to be pathogenic from 3/5 pathogenicity algorithms were retained *(n* = 93) (Appendix A). Candidate genes affected by these variants in individual families are listed in Table 2. Furthermore, we detected 143 variants in 66 cancer-associated genes, including *BMP5*, with the highest predicted deleteriousness score of 22.7, *MET* and *USP44* (Appendix A). None of the detected variants in cancer genes were officially classified as inherited germline disease-causing mutations (HGMD^®^ Professional 2020.3).

Subsequently, we investigated genes that were recurrently hit in at least two families, which could point to a molecular mechanism of pathogenesis for AGCT. Across all coding genes, we identified fourteen genes that were affected by identical or different single-nucleotide variants in the same gene among at least two families (Appendix A). No genes remained after filtering out variants present in low quality or intronic regions and genes that commonly have false-positive variant calls from next-generation sequencing.

### 3.3. Whole-Genome Analysis

Genome wide, we found 13,543 single nucleotide variants present in at least 1 family with a CADD PHRED deleteriousness score of ≥5 (Figure 2 and Appendix A). Variants with the highest deleteriousness score (CADD PHRED > 30) included frameshift variants predicted to result in loss of function of *PIK3C2G* (p.Asn1129Thrfs*27), *KLHL33*, and *MYH1* (rs545765873), and three intergenic variants. Furthermore, 52 single nucleotide variants were shared by at least two families. Two single nucleotide variants were present in three families, of which one could not be validated by PCR and was classified as a sequencing artefact (rs1306359244, chr12:103223803, near *IGF-1*). The other variant (rs781644268, chr12:53833990) was an intergenic variant downstream of *AMHR2* and upstream of *PRR13* and positioned at a long T stretch.

A total of 563 structural variants were identified that were shared within at least one family, including 72 structural variants shared by two different families (Figure 2 and Appendix A). These structural variants collectively contained 998 individual genes that were affected by these variants. We assessed the genes that were affected in at least two families *(n* = 104; Appendix A). No gene was affected in more than two families.

AGCTs have characteristic somatic copy number gains and losses that are present in a large proportion of tumors, specifically gain of chromosome 14 and concurrent loss of chromosome 22 [14]. It is unclear whether these structural variants facilitate tumor development, or if also germline copy number variation could contribute to AGCT evolution. We did not identify germline duplications on chromosome 14 or loss on chromosome 22 affecting multiple families.

### 3.4. Variants in Genes Associated with Sex Cord-Stromal Tumor or Hereditary Ovarian Cancer 

Genes associated with the development of sex cord-stromal tumors or hereditary ovarian cancer were specifically investigated. No rare (<1:1000) germline single nucleotide or structural variants were identified within 75 kB of *FOXL2, STK11/LKB1, IDH1/2, DICER1,* or for *BRCA2*. A noncoding mutation shared by five individuals across three families (rs12938971) was identified 22 kb downstream of the coding region of *BRCA1,* however this variant was present in a polyG stretch in a low complexity region and, therefore, likely to be a sequencing artefact. One report showed an AGCT in both a *BRCA1* and a *BRCA2* series of ovarian cancers. That study investigated individuals with breast and/or ovarian cancer, from families with at least one ovarian cancer diagnosis [27].

### 3.5. Comparison with Unrelated AGCT Patients

We compared the clinical characteristics and germline variants of the patients from the four families to a reference group of 33 unrelated AGCT patients. The mean age of the familial patients at diagnosis was 38 years (range from 17 to 60), significantly lower than the reference group (mean age at diagnosis 51 years, ranging from 29 to 75, T-test *p*-value = 0.016), and seven of the eight women were premenopausal at the time of diagnosis. Although one of the patients had metastatic disease at diagnosis (12.5% vs. 3% in the reference group), none of the patients developed a recurrence. The genomic relatedness within the unrelated AGCT patient cohort was predicted to be at least 7th degree and between the families and the unrelated patients at least 8th degree (Appendix A), indicating no relationship between these patients.

Furthermore, we investigated if recurrent or predicted pathogenic single nucleotide variants and structural variants were present in a reference group of unrelated AGCT patients for whom whole-genome sequencing had been performed on germline DNA derived from blood or saliva [14]. Of the 52 single nucleotide variants shared by at least two families in the present study, 26 variants were present in the reference group and eight variants present at least two times in the reference group. No variant was in a coding area of the genome, and manual inspection of the next-generation sequencing reads of the one variant (rs1185417161) seen seven times in our reference AGCT group revealed reads with many mismatches and most likely mapped to the genome incorrectly. Finally, no structural variants present in this study were identified in the reference group. None of the patients in the reference group had a *PIK3C2G* variant and one patient had a *BMP5* variant, although it was not predicted to be pathogenic. A predicted damaging *LRP2* p. His896Gln variant was identified in family A. Homozygous pathogenic variants in *LRP2* result in Donnai–Barrow syndrome [28], and heterozygous loss-of-function variants are not well tolerated in the general population [29]. Splice variant *LRP2* somatic mutations have also been identified previously in AGCTs [30]. We identified thirteen exonic nonsynonymous *LRP2* variants in the reference group, of which two were predicted to be pathogenic by 3/5 prediction algorithms (Appendix A). One of these variants was identified only once (rs760331558) and the other (rs61995913) was present at an allele frequency of 0.004. Therefore, no specific mutations were shared, although predicted pathogenic variants in *LRP2* were found in both related patients and the reference group.

## 4. Discussion

This study describes the familial occurrence of adult granulosa cell tumors of the ovary. For the first time, the germline DNA of familial AGCT patients was investigated, with the aim to identify an overlapping genetic variant or affected gene that could have contributed to tumor evolvement. The genomic analyses covered both small single nucleotide variants, insertions and deletions, and larger structural variation. We focused on both variation that was shared between families as well as predicted damaging variants present in one family. 

Although we did not find a genetic variant that was shared within all four families, we did identify variants in genes that were predicted to be damaging. Of these genes, *LRP2* (variant in family A) is involved in lysosomal regulation of lipid metabolism [31]. Additionally, a nonsynonymous coding variant in *LRP1* was detected in family C. Although this variant was not predicted to be damaging, *LRP1* is also involved in lipid metabolism [31]. Previous studies have shown that the lipid metabolism in granulosa cells plays a vital role during follicular development and is indispensable for oocyte maturation [32,33,34]. In addition, *LRP2* expression is induced by the peroxisome proliferator-activated receptor-gamma (PPARγ), a key transcriptional factor regulating lipid metabolism which is widely expressed in granulosa cells [35]. PPARγ activation, combined with inhibition of the X-linked inhibitor of apoptosis protein (XIAP), has been suggested as a novel therapeutic strategy in AGCTs [36]. 

Furthermore, we found a predicted damaging *PIK3C2G* variant in family C (deleteriousness score: 33, predicted to result in nonsense-mediated decay). *PIK3C2G* is involved in cell signaling pathways that regulate cell proliferation, survival, and oncogenic transformation, and is altered in 0.58% of all cancers [37]. An analysis of 4034 cases from The Cancer Genome Atlas identified germline truncating mutations in 34 genes, including *PIK3C2G.* Germline *PIK3C2G* truncating variants were associated with cancer predisposition, specifically with ovarian cancer [38].

We also found a very rare predicted damaging heterozygous variant in *BMP5* in family B. Bone morphogenetic proteins (BMPs) play an important role in embryonic and postnatal development by regulating cell differentiation, proliferation, and survival, thus maintaining homeostasis during organ and tissue development [39]. BMPs can serve as either oncogenes or tumor suppressors, leading to tumorigenesis and regulating cancer progression [40]. The BMPs are cytokines belonging to the Transforming Growth Factor (TGF)-β superfamily, which also includes TGF-βs, activin, inhibin, nodal, and myostatin [39]. BMPs activate SMAD pathways, phosphatidylinositol 3-kinase (PI3K)/AKT, mitogen-activated protein kinase (MAPK), nuclear factor kappa B (NF-κB), and Janus kinase/signal transducer and activator of transcription (JAK/STAT) signaling pathways. A recent study found that TGF-β signaling enhances the effect of mutant *FOXL2* (c.402C > G) and BMP stimulation (via signaling through *SMAD1*, *SMAD5* and *SMAD8*) seems to counteract this effect. Moreover, conditional double deletion of *SMAD1* and *SMAD5* or triple deletion of *SMAD1, SMAD5, SMAD8* led to infertility and granulosa cell tumor development in mice [41]. *BMP5* somatic missense mutations are present in 7.7% of colorectal cases and reduction in BMP5 through loss of function or damaging nonsynonymous variants has been linked to disease progression in cancer [42,43]. Targeting BMPs and their receptors were successful in preventing tumor growth and invasion in preclinical and clinical cancer studies [44]. 

It is not yet known whether a genetic cause for AGCT exists in these patients, since we could not identify a shared variant or affected locus between the families that could be linked to granulosa cell tumor development. Additionally, the families included many unaffected females, indicating that if an autosomal dominant causal variant is present, it could be a variant with incomplete penetrance. Reduced penetrance may result from a combination of genetic, environmental, and lifestyle factors. This phenomenon can make it challenging for genetic professionals to interpret an individual’s family medical history and predict the risk of passing on a genetic condition. However, affected family members from both family A and D are fifth-degree relatives and their environmental and lifestyle factors are therefore likely to differ. Another possibility is a familial detrimental variant in a tumor suppressor gene or a gene involved in granulosa cell proliferation, in which another somatically acquired variant is needed to knock out or reduce function of the gene and initiate tumorigenesis or facilitate tumor development.

However, the identification of four different families, of which three families originate from a country with only 15–20 new cases per year, including first-, second- or fifth-degree relatives with a very rare gynecological tumor makes a genetic germline contribution plausible. The fact that the age of onset in the familial cases was significantly lower than in incidental AGCTs, with the youngest patient being diagnosed at the age of 17 years, strengthens this hypothesis as a disease caused by a germline variant usually manifests at a younger age.

Besides the occurrence of granulosa cell tumors, cases of breast cancer were found in first-degree relatives of the AGCT patients in two families. In these patients, the median age at diagnosis was lower (51.7 years) than the average age at breast cancer diagnosis in the general population (62 years) [45]. A potential link between AGCTs and breast cancer has previously been suggested. A recent study including 1908 AGCT cases found a higher observed (*n* = 79, 4.14%) than expected number of breast cancer cases (*n* = 27, 1.41%) [46]. Moreover, other studies report a breast cancer rate of 5–10% among AGCT patients [47,48,49]. Furthermore, two AGCT patients were subfertile and diagnosed with polycystic ovary syndrome. Several studies reported a possible connection of granulosa cell tumors to subfertility, with improved fertility after surgical removal of the tumor [50]. Granulosa cell tumors at the fertile age were associated with nulliparity and with a clinical presentation of anovulatory infertility, while AGCTs later in life were associated with a normal average fertility pattern. In our study, breast cancer and subfertility were also present in non-affected family members. The co-occurrence of granulosa cell tumors, breast cancer, and/or subfertility in these families may indicate a shared etiology, for example, a genetic predisposition.

It is known that trying to resolve a familial disease can be a daunting task. For example, in familial high grade ovarian carcinoma, the hereditary basis of approximately 50% of cases is still unexplained. A recent study on suggested familial BRCA1/2 wildtype high grade ovarian cancers found that only 6.6% of cases could potentially be explained by genes known or suggested to be linked with a higher risk of ovarian cancer [51]. This study reported a high number of individual rare loss of function variants, suggesting that these could be genuine predisposing variants, which is in agreement with our findings (rare loss of function variants in *PIK3C2G*, *KLHL33*, and *MYH1*).

The ongoing advances in the field of whole-genome sequencing data analysis may lead to novel insights and may resolve unexplained familial cancer cases. Nonetheless, the identification of a causal germline variant may not have direct clinical implications as, due to the rarity of the disease, genetic testing for AGCT patients or their family members in general may not be necessary. Moreover, the identification of four families and one previously reported family indicates that the vast majority of cases are sporadic rather than familial. However, identification of a predisposing genetic factor could help to unravel the pathological mechanism of AGCTs.

## 5. Conclusions

Adult granulosa cell tumors can occur in familial clusters and could potentially be hereditary. We did not identify an overlapping genetic variant or affected genetic locus that may explain a genetic predisposition for AGCT in the four investigated families. In these families, the age of AGCT diagnosis was lower than in unrelated cases, and breast cancer, PCOS, and subfertility were reported in these families, suggesting potential shared etiologic factors.

## Figures and Tables

**Figure 1 cancers-13-02430-f001:**
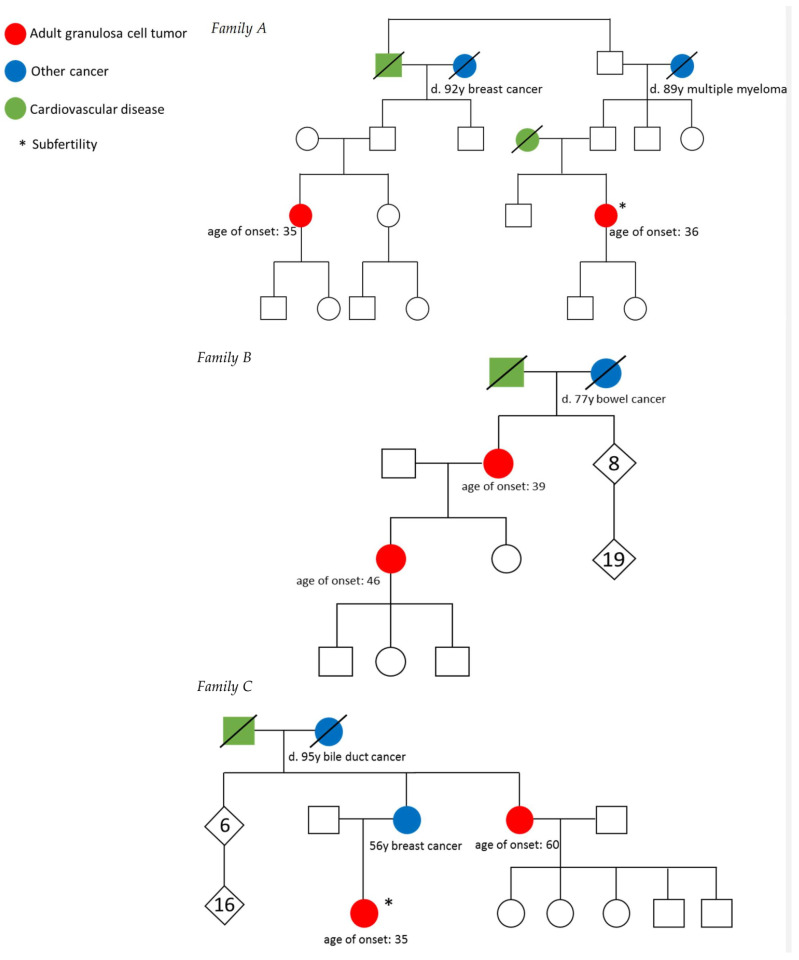
Family pedigrees. Colors represent diagnosis of AGCT, other cancer or cardiovascular disease. Deceased persons are indicated with a slash (/). Family A. Patients were related via their grandfathers. Family B. The AGCT patients represented mother and daughter. Family C. The AGCT patients represented niece and aunt. Their mother, respectively sister was diagnosed with breast cancer. Patient C2 was involuntarily childless because of subfertility and more cases of subfertility were reported in this family. Family D. The AGCT patients were related through their grandfathers, who were brothers. Both mothers of the patients were diagnosed with breast cancer. One of the patient’s sisters was also involuntarily childless due to subfertility.

**Figure 2 cancers-13-02430-f002:**
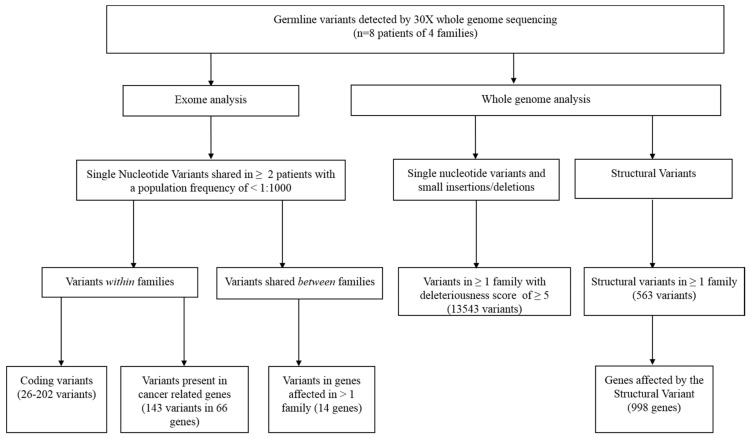
Filter approach.

**Table 1 cancers-13-02430-t001:** Patient characteristics.

Patient	Age at Diagnosis	Tumor Stage	Treatment **	Medical History	Family History
A1	35	IIB *	Surgery	Clubfoot	
A2	36	IA	Surgery	Polycystic ovary syndrome, 2x vaginal delivery after in vitro fertilization	
B1	46	IC	Surgery		
B2	39	Unknown	Surgery and radiotherapy		
C1	35	IA	Surgery	Polycystic ovary syndrome, subfertility	Breast cancer, PCOS, subfertility
C2	60	IA	Surgery		
D1	17	IC	Surgery and chemotherapy		Breast cancer, subfertility
D2	39	IC	Surgery and chemotherapy		

* location dorsal from the uterus. ** no recurrences occurred, and all patients were currently alive with no evidence of disease.

**Table 2 cancers-13-02430-t002:** Detected overlapping germline variants per family.

Family	Gene	Effect	Variant (cDNA)	Variant (Protein)
A	*HTRA4*	Nonsynonymous	c.1009G > C	p.V337L
	*LRP2*	Nonsynonymous	c.2688C > G	p.H896Q
	*PCSK9*	Nonsynonymous	c.479G > A	p.R160Q
B	*BMP5*	Nonsynonymous	c.1291T > C	p.Y431H
	*CRLF2*	Frameshift	c.496_497delGT	p.N166Yfs*111
	*FSCN3*	Nonsynonymous	c.212G > A	p.G71D
	*HFM1*	Nonsynonymous	c.4283T > C	p.L1428S
	*MET*	Nonsynonymous	c.3409G > A	p.G1137R
	*NOX5*	Nonsynonymous	c.1600G > C	p.G534R
	*SPTBN5*	Nonsynonymous	c.10672T > C	p.W3558R
	*TEAD2*	Frameshift	c.1286_1287delAT	p.Y429Cfs*55
C	*CBX8*	Nonsynonymous	c.916C > T	p.R306W
	*HYDIN*	Frameshift	c.6584_6585ins59	p.P2196Ifs*17
	*IGSF1*	Nonsynonymous	c.709C > G	p.P237A
	*LAMA3*	Nonsynonymous	c.701T > A	p.I234K
	*PSMD5*	Nonsynonymous	c.820G > A	p.V274M
	*PXDN*	Nonsynonymous	c.3464C > A	p.A1155E
	*TBP*	Frameshift	c.231_234delGCAGinsCAG	p.Q77Hfs*67
D	*USP44*	Nonsynonymous	c.1250G > A	p.R417H
	*RASSF2**	Nonsynonymous	c.389T > A	p.L130Q

Reported variants included exonic variants passing quality filters that resulting in a frameshift, stop/start loss, or a nonsynonymous variant predicted to be pathogenic from 3/5 pathogenicity algorithms and candidate genes affected in two families. * Variant did not meet inclusion criteria but was reported as the gene is a tumor suppressor, and the variant is rare and was predicted damaging in 2/5 algorithms.

## Data Availability

WGS Binary Alignment Map (BAM) files are available through controlled access at the European Genome-phenome Archive (EGA), hosted at the EBI and the CRG (https://ega-archive.org, 8 January 2021), with EGA dataset ID: EGAD00001006780. Requests for data access will be evaluated by the UMCU Department of Genetics Data Access Committee (EGAC00001000432) and transferred on completion of a material transfer agreement and authorization by the medical ethical committee of the UMCU to ensure compliance with the Dutch ‘medical research involving human subjects’ act.

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
