# Peer review of "Familial Occurrence of Adult Granulosa Cell Tumors: Analysis of Whole-Genome Germline Variants"

_cancers, 2021, doi:10.3390/cancers13102430_

Round 1
Reviewer 1 Report
The authors report a series of 4 families with familial occurrence of adult granulosa cell tumor. Although even with a lifetime risk of about 1 in 1400 (5% of 1 in 70) occasional families will have two cases by chance, these families are plausible for inherited predisposition with reduced penetrance due to earlier onset and that 3 occurred in a country with a relatively small population. The authors undertook genome sequencing and found possible deleterious variants in PIK3C2G, BMP5 and LRP2, but no gene was shared between any two families. This is an interesting report that is generally well written.
Specific points
- ‘Germline mutations are involved in the evolution of (some) ovarian cancers’ -please qualify this statement by adding ‘some’
- ‘This study reported a high number of individual (with) rare loss of function variants, suggesting that this (these) could be genuine predisposing variants, which is in agreement with our findings (rare loss of function variants in PIK3C2G, KLHL33 and MYH1). -Please change sentence for sense
- Was a comprehensive search made for BRCA1/2 variants and MLPA undertaken? Two of the families had potential ‘obligate’ carriers with early onset breast cancer.
- Please reference any known attempts to assess BRCA1/2 in AGST. One report showed an AGCST in both a BRCA1 and a BRCA2 series of ovarian cancers-https://pubmed.ncbi.nlm.nih.gov/18312450/
Author Response
We would like to thank the reviewer for his/her time to review this manuscript.
- We have added ‘some’ in line 56.
- We have modified line 446.
- Yes, we specifically investigated BRCA1 and 2 and checked for:
- Coding variants and predicted damaging variants
- Overlapping variants
- Noncoding variants present in >1 patient with a population frequency of <1:1000
- We also visually checked BRCA1 and 2 in IGV for variants
- We checked whether SV breakpoints were located in BRCA1 or 2
We performed this analysis for all known ovarian cancer related genes (TP53, BRCA1/2, TERT, FOXL2, RAD51C RAD51D BRIP1 PALB2 and ATM). No predicted pathogenic variants were detected or overlapping variants or coding variants with a population frequency of <1:1000.
We have attached a file with all variants in ovarian cancer related genes with reported allele frequency and predicted pathogenicity score (CADD).
- Many thanks for this suggestion. We have added this study in line 323:
“One report showed an AGCST in both a BRCA1 and a BRCA2 series of ovarian cancers. That study investigated individuals with breast and/or ovarian cancer, from families with at least one ovarian cancer diagnosis.” To our knowledge, no additional studies exist that report on familial germline BRCA variants in AGCT.
Reviewer 2 Report
This is a good preliminary investigation of potential genetic links to a rare gyn malignancy. The number of subjects studied is small as would be expected for a rare disease. The paper is well written and the conclusions seem valid to me. If there are issues with the small study population the paper might be called a pilot study. Otherwise I think it is a publishable paper as a first step, albeit negative, in learning the genetic etiology of a rare human tumor.
Author Response
We would like to thank the reviewer for his/her time to review this manuscript. Our research group collaborates with patients organizations internationally and via this organization we will keep searching for familial cases. If additional cases present in the future, we will continue to collect and analyze data.